# Effects of Dietary Oregano Essential Oil on Cecal Microorganisms and Muscle Fatty Acids of Luhua Chickens

**DOI:** 10.3390/ani12223215

**Published:** 2022-11-20

**Authors:** Tao Wu, Farong Yang, Ting Jiao, Shengguo Zhao

**Affiliations:** 1College of Animal Science and Technology, Gansu Agricultural University, Lanzhou 730070, China; 2Animal Husbandry Pasture and Green Agriculture Institute, Gansu Academy of Agricultural Sciences, Lanzhou 730070, China; 3College of Grassland Science, Gansu Agricultural University, Lanzhou 730070, China

**Keywords:** oregano essential oil, fatty acid, 16S rRNA, microorganisms, Luhua chickens

## Abstract

**Simple Summary:**

As a natural substitute for antibiotics, oregano essential oil is a feed additive drug approved by the Ministry of Agriculture, which can improve animal production and health. There are few reports on the combination of quinoa and oregano essential oil used in animal production for feed development and utilization. Therefore, the aim of this study was to analyze the effects of different proportions of oregano essential oil in a corn–quinoa–soybean meal diet on the intestinal microbiota and muscle fatty acids of Luhua chickens. The results showed that the addition of oregano essential oil to the corn–quinoa–soybean meal diet effectively improved the microbial community composition in the caecum of Luhua chickens, enriching the related pathways, so as to promote the digestion and absorption of nutrients and enhance intestinal barrier functioning. It significantly improved the content of some fatty acids in the muscles of Luhua chickens, and there was a certain correlation between cecal microorganisms and fatty acid deposition in muscles. These data provide a theoretical basis for the scientific application of oregano essential oil and quinoa in the feeding and fodder development of Luhua chickens.

**Abstract:**

This experiment was conducted to investigate the effects of oregano essential oil on the cecal microorganisms and muscle fatty acids of Luhua chickens. One hundred and twenty 49-day-old healthy dewormed Luhua chickens were randomly divided into four groups with three replicates per group and ten chickens per replicate. The corn–quinoa and soybean meal diets were supplemented with 0 (Q8 group), 50 (QO50 group), 100 (QO100 group) and 150 mg·kg^−1^ (QO150 group) of oregano essential oil, respectively, and the experiment lasted for 75 days. The composition of intestinal flora was detected by Illumina sequencing of the 16S rRNA V4 region, and the composition and content of fatty acids in the muscles were analyzed by gas chromatography. The results showed that dietary oregano essential oil can effectively increase the contents of elaidic acid (C18:ln9t), polyunsaturated fatty acids (PUFAs) and n-3 polyunsaturated fatty acids (n-3 PUFAs) in breast muscle tissues. However, the fatty acid composition and PUFA content in leg muscle tissues were not significantly improved. According to a 16S rRNA high-throughput sequencing analysis, dietary oregano essential oil supplementation with a certain concentration can change the cecal microbial community composition of broilers. At the phylum level, *Elusimicrobia* in the QO150 group was significantly lower than that in Q8 group (*p* < 0.05). At the genus level, *Phascolarctobacterium*, *Parasutterella* and *Bilophila* in the experimental groups (QO50, QO100 and QO150) were significantly lower than those in the Q8 group (*p* < 0.05). An enrichment analysis of the microbial function found that the amino acid metabolism, energy metabolism, metabolism, signal transduction and genetic information processing were mainly enriched in the experimental groups, which promoted the digestion and absorption of nutrients and enhanced intestinal barrier functioning. An analysis of the association between fatty acids and microbes found that the abundance of microbiota was significantly correlated with partially saturated fatty acids (SFAs) and unsaturated fatty acids (UFAs) (*p* < 0.05). In conclusion, the dietary addition of oregano essential oil can effectively improve cecal microbial community composition, promote the digestion and absorption of nutrients, and enhance intestinal barrier functioning. It can significantly improve the content of some fatty acids, and there was a certain correlation between caecum microorganisms and fatty acid deposition in muscles.

## 1. Introduction

In recent years, the use of antibiotics in feed has seriously affected the health and ecological environment of animals and people. At present, green and healthy breeding has become a key concern in the development of animal husbandry. Therefore, it is urgent to find green, safe and efficient substitutes for antibiotics in feed. In the past few years, as a potentially useful source of bioactivity, essential oils have received particular attention, and as natural agents, plant extracts have been used to replace antibiotics in poultry diets to improve the safety of by-products [1,2]. Oregano essential oil is a volatile plant essential oil extracted from oregano. The main chemical components of oregano essential oil are volatile phenolic compounds (carvatol and thymol), which are antioxidant, anti-inflammatory, antibacterial and antiviral, while enhancing immunity and performance in animals, etc. [3,4,5]. Oregano essential oil is environmentally friendly, safe and free of drug resistance and toxic side effects. It has gradually become a substitute for antibiotics and plays an important role in modern livestock and poultry production [6,7,8]. In recent years, scholars at home and abroad have studied the application of oregano essential oil in animal husbandry.

It is well known that the diversity of intestinal microbiota plays an important role in host metabolism, nutrient digestion, growth performance and health [9,10]. Studies have shown that the relative abundance of *Clostridium sensu Stricto_1* and *Lactobacillus genera* in broilers increases when oregano essential oil is added to their diet [11]. Bauer et al. [12] showed that dietary supplementation with 1% and 2% oregano could reduce the abundance of *Proteus jejuni*, *Klebsiella*, *Staphylococcus* and *bifidobacteria* in Ross308 broilers. The cecum of chickens is the most important distal intestinal segment, and the intestinal microbe concentration is the highest in adult chickens, which affects their health and growth performance [13,14]. The cecum is a complex ecosystem consisting of a highly diverse microbiome. In recent years, many studies have used high-throughput sequencing technology to study cecal microbial diversity. The digestion and absorption of the cecum is related to cecal microorganisms [15]. Compared with normal poultry, cecectomized poultry have a lower digestibility and metabolic capacity for crude fibers or other nutrients [16]. Therefore, the cecal microbiota has attracted extensive attention [14,17]. Analyses of cecal microbiotas are a key area of poultry nutrition research, which will contribute to a further understanding of microbial diversity and host interactions [18]. Microorganisms in the cecum actively ferment and convert the fiber contents into digestible components of the animal host. The cecum is a complex ecosystem that hosts a wide variety of microbiota and is an important factor in animal production. However, there are few studies on the cecal microbial diversity of Luhua chickens. Therefore, it is necessary to study the intestinal microbial diversity of Luhua chickens. Studies have found that there is a certain correlation between the cecal microbial community and the meat quality of broilers [19].

With the development of society and improvements in people’s living standards, meat plays an important role in the human diet. The poultry meat can provide abundant fatty acids, protein, minerals and trace elements, so the modern poultry industry has been paying more and more attention to improving poultry production performance and meat quality to provide nutritious meat to an overwhelming number of customers. Relevant studies have shown that a high concentration of n-3 PUFAs could increase the concentration ratio of unsaturated fatty acids (PUFA/SFA) and n-3/n-6, which are beneficial to human health [20]. Due to the low intake of n-3 PUFAs in the human body [21], scientists have shown great research interest in producing chickens rich in n-3 PUFAs for human consumption [22]. Oregano essential oil has an important effect on the quality of animal products. Studies have shown that adding oregano essential oil to the feed can promote the feed conversion rate as well as improve the ketone body quality and meat quality of broilers [23]. Adding oregano essential oil to diets can improve the yield and fat content of milk [24]. Dietary oregano essential oil significantly reduces the content of SFAs and significantly increases the content of UFAs in beef. The composition and content of fatty acids in muscles affect the nutritional value and flavor of the meat.

So far, no studies have evaluated the effects of oregano essential oil supplementation on the intestinal microflora and muscle fatty acids of Luhua chickens. In this study, 16S rRNA microbiome and gas chromatography were used to evaluate the effects of oregano essential oil with three different gradients on the intestinal microbiota and fatty acids of Luhua chickens fed quinoa as a raw material. A correlation analysis between the gut microbes and fatty acids can provide a theoretical basis for exploring the mechanism of action of oregano essential oil on the cecal microbiota and fatty acid deposition of broilers, as well as insight into its application in Luhua chicken production.

## 2. Materials and Methods

### 2.1. Moral Statement

All procedures for using experimental animals have been approved by the Experimental Animal Management Committee of Gansu Agricultural University with license number GSAU-2019-0018, and the animals were tested in accordance with the guidelines of the Committee. Experiments involving animals were conducted in accordance with the Regulations on the Management of Experimental Animals (Ministry of Science and Technology; revised June 2004). The samples were collected in accordance with the Guidelines of Ethics Committee for the Care and Use of Experimental Animals at Gansu Agricultural University.

### 2.2. Experimental Design

A total of one hundred and twenty 49-day-old dewormed healthy female Luhua chickens (body weight 1478.75 ± 85.81 g) were randomly divided into 4 treatment groups with 3 replicates per treatment and 10 chickens per replicate. Broilers in control group (Q8 group) were fed a basal diet, and broilers in experimental groups (QO50, QO100 and QO150 groups) were fed the experimental diets supplemented with 50, 100 and 150 mg·kg^−1^ oregano essential oil, respectively.

### 2.3. Experimental Diet and Feeding Management

#### 2.3.1. Diet Composition and Nutrient Level

The diets of experimental chickens were formulated according to the Chinese Feeding Standard of Chicken (NY/T33-2004), and the nutrient levels all met the nutritional needs of Luhua chickens. The composition and nutrient levels of experimental diets are shown in Table 1. Oregano essential oil was purchased from Reco Company in the United States. The purity of oregano essential oil was 1.13% and the carrier content was 98.87%. The oregano essential oil carrier is composed of 75% clinoptilolite, 13.87% limestone and 10% diatomite. Quinoa was provided by Tianzhu Quinoa Alpine Experimental Station, Gansu Academy of Agricultural Sciences. The basal diet was produced by Gansu Aonong Feed Science and Technology Co., LTD., according to the experimental formula.

#### 2.3.2. Experimental Animal Feeding and Management

Ten days before the experiment, the coop was thoroughly cleaned and washed, the utensils, such as the sink and trough, were cleaned, and the walls were also disinfected with lime water. After cleaning and drying, we closed and sealed the doors and windows of the henhouse as well as the ventilation fan, fumigated the whole henhouse with potassium permanganate for 24 h, and then opened the doors and windows for ventilation for 1 week. The coop was kept dry, clean and well ventilated. The chickens were examined, immunized, disinfected, numbered and weighed. The chickens from the farm were sterilized before being transferred in the henhouse. Spirit (powder, water spray) in proportion was used to disinfect the transport vehicles to the chicken coop. During the breeding period, the coop (with chickens) once every 2 weeks with sanitizer or 0.1% Germicide, waste was regularly cleaned from the henhouse as much as possible and steps were taken to avoid inducing various chicken disease. During the experiment, the chickens were fed twice at fixed time (07:00 and 15:00) every day, and were free to eat and drink. We added sufficient feed to each trough to ensure that there was plenty when the chickens were eating. We regularly weighed and recycled the leftovers. The chickens were also regularly weighed. The feeding experiment was carried out in the training base of Gansu Agricultural University, and the diet was in the form of a powder during the whole experiment period. During the rearing process of broilers, the environmental indicators of the henhouse were strictly controlled and recorded. The control range of the main indicators were as follows: temperature was 22–28 °C, humidity was 60–65%, light was time-controlled at about 16 h (lights on from 6:00 a.m. to dawn, light on at 18:00 p.m. and lights off at 22:00), and the coop was divided into three layers: upper, middle and lower. The experimental period lasted for 75 days, including the pre-experimental period of 3 days and the formal experimental period of 72 days.

### 2.4. Collection and Treatment of Test Samples

#### 2.4.1. Collection and Processing of Muscle Tissue Samples

On the 75th day of the experiment, two Luhua chickens were randomly selected from different replicates in each group. After weighing them live, the chickens were bled to death by neck, their feathers were removed, their water was drained and they were slaughtered for measurements. Breast muscle and leg muscle samples with skin and fascia removed were collected, vacuously packed and placed in a refrigerator at −80 °C for testing.

#### 2.4.2. Collection of Cecal Microbial Samples

After 121 days, two Luhua chickens were randomly selected from different replicates in each group and slaughtered, with six in each group. Sterilized surgical scissors were used to separate the cecum and other intestinal segments, and the cecum contents were divided into 5 mL cryopreserved tubes and quickly put into a cryogenic liquid nitrogen tank at −80 °C to save the contents for measuring intestinal microbial diversity. 

### 2.5. Experimental Methods

#### 2.5.1. Determination of Fatty Acid Content in Muscle

The meat samples in the vacuum packaging bag were thawed at room temperature for 12 h, and the muscle surface was peeled off and removed with a knife. The samples were placed in a mortar and ground with liquid nitrogen, and then 1.0 g of each ground-up sample was weighed in a 10 mL stopper tube. Volumes of 0.7 mL of KOH solution at a concentration of 10 mol·L^−1^ and 5.3 mL of anhydrous methanol (analytical methanol (for chromatography)) were added, respectively, and the test tube was shaken for 5 s every 20 min in a water bath at 55 °C for 1.5 h. At the end of the water bath, the tube was removed and cooled to below room temperature under tap water. Then 0.58 mL of 12 mol∙L^−1^ H_2_SO_4_ solution was added, and the constant-temperature water bath continued for 1.5 h at 55 °C for the methylation of free fatty acids, while the tube was shaken every 20 min for 5 s. At the end of the water bath, we took out the test tube, cooled it with tap water to below room temperature, added 3 mL n-hexane and shook the mixture well. Then we transferred it to a centrifuge tube, centrifuged it at 3000 r·min^−1^ for 5 min, took the supernatant and filtered it into the sample bottle using organic-phase filtration membrane, and heated it at 45 °C to concentrate each 2 mL sample to less than 1.5 mL. We set the temperature at −20 °C for GC detection.

Three parallel samples were tested for the same sample, and the average value of the three samples was taken as the test result of the sample. Fatty acids were identified according to the relative retention times of fatty acid methyl ester standards. Using the peak of the standard sample (Figure 1) and comparisons with the sample peak diagram, a variety of fatty acids were compared, and the peak area normalization method was used to determine the relative percentage of each fatty acid, and finally the average percentage of each fatty acid. The ratio of PUFAs/SFAs to n-6/n-3 was calculated according to the composition and specific content of the measured fatty acids.

##### Chromatographic Condition

The contents of fatty acids in muscle were determined by gas chromatography (GC-2010 plus; Shimadzu Company, Japan). The determination was performed on an SPTM-2560 capillary column (100 m × 0.25 mm × 0.2 m) with sample size of 1.0 μL. Chromatographic conditions were as follows: inlet detector temperature was 250 °C, nitrogen flow rate was 1.11 mL·min^−1^, and split ratio was 100:1. Programmed temperature mode was as follow: the initial temperature was kept at 100 °C for 5 min, and then it was increased to 240 °C at 4 °C·min^−1^ for 30 min. The total duration was 70 min.

#### 2.5.2. Total DNA Extraction, PCR Amplification and 16S rRNA High−Throughput Sequencing

##### Extraction of Total DNA

Cecal contents were collected from 6 chickens in each group for intestinal microbiome analysis. Total DNA was extracted by CTAB method, and its concentration and purity were detected by Nanodrop 2000 ultra-fine spectrophotometer. After that, agarose gel electrophoresis was used for detection. Appropriate amount of DNA was taken into the centrifuge tube, and the samples were diluted to 1 ng·μL^−1^ with sterile water. DNA samples were stored at −80 °C for later use.

##### PCR Amplification

Using the diluted total DNA as a template, the cecal microflora was characterized by amplification of V4 region of 16S rRNA gene according to the selection of sequencing regions. Then, specific primers with barcodes (515F and 806R) were used to identify bacterial diversity in 16S V4 region. The specific primers were 515F (5’-GTGCCAGCMGCCGCGGTAA-3’) and 806R (5’-GGACTACHVGGGTWTCTAAT-3’). Phusion^®^ High-Fidelity PCR Master Mix with GC Buffer from New England Biolabs was used for PCR to ensure amplification efficiency and accuracy.

Reaction procedure was as follows: pre-denaturation was performed at 95 °C for 5 min and 25 cycles (95 °C denaturation for 30 s, annealing at 55 °C for 30 s, extended 40 s at 72 °C). Extension was performed at 72 °C for 7 min.

##### Mixing and Purification of PCR Products

The PCR products were detected by electrophoresis using 2% agarose gel, and the same amount of samples was mixed according to the concentration of PCR products. The PCR products were detected by electrophoresis using 2% agarose gel after being fully mixed. For the target strip, the adhesive recovery kit provided by Qiagen Company was used to recover the product.

##### Library Construction and Computer Sequencing

TruSeq^®^ DNA PCR-Free Sample Preparation Kit was used for library construction. The constructed libraries were quantified by Qubit and Q-PCR. After the qualified libraries were tested, NovaSeq6000 (Illumina, San Diego, CA, USA) was used to finish the sequencing.

#### 2.5.3. Bioinformatics Analysis

The data obtained from sequencing on Illumina NovaSeq platform were spliced with various sample reads using FLASH (V1.2.7) [25] to obtain raw tags; then, after strict filtering [26], the tags data (clean tags) were obtained. The tags were then compared with the species annotation database to remove the chimera sequences and obtain effective tags [27]. The effective tags were clustered by operational taxonomic units (OTUs) with 97% similarity using Usearch software (Uparse v7.0.1001). The OTUs were annotated based on Silva (bacterial) taxonomy database. Based on the results of OTUs analysis, the samples were analyzed at each taxonomic level, and the community composition of each sample was obtained at the taxonomic levels of kingdom, phylum, class, order, family, genus and species. R software (Version 2.15.3) was used for Alpha diversity analysis and Beta diversity index difference analysis between groups to obtain Alpha diversity index Ace, Chao1, Shannon, Simpson index and sample dilution curve. PICRUST was used for functional prediction of microbial communities in samples [28,29,30].

#### 2.5.4. Statistical Analysis

All statistical analyses were performed using SPSS 22.0 software (SPSS Inc, Chicago, IL, USA). The difference in fatty acid compositions in breast and leg muscles was analyzed, and the results were expressed as mean ± standard deviation. *p* < 0.05 was considered statistically significant, and *p* < 0.01 was considered extremely significant. Spearman correlation test was used to analyze the correlation between fatty acid composition and pectoral muscle, leg muscle and cecal microbe contents (relative abundance > 0.5%).

## 3. Results

### 3.1. Effects of Dietary Oregano Essential Oil on Fatty Acids in Breast Muscle of Luhua Chickens

Twenty-four fatty acids were detected in the muscles of each group (Table 2), including eight kinds of SFAs and sixteen kinds of UFAs. Among the UFAs, there were seven monounsaturated fatty acids (MUFAs) and nine PUFAs. The content of C18:1n9c was the highest with about 32%, followed by C16:0 with about 24%, and C18:2n6c was the third highest with about 13%. As can be seen from the table, there were significant differences in some pectoralis fatty acids in the diets supplemented with a certain amount of oregano essential oil, and the contents of C4:0, C18:1n9t and C20:5n3 (EPA) in the experimental groups (QO50, QO100 and QO150) were significantly higher than those in the Q8 group (*p* < 0.05). The content of C14:1 and the PUFA/SFA ratio in the QO100 group were significantly higher than those in the other experimental groups and the Q8 groups (*p* < 0.05). The contents of C18:2n6t and C20:2 in the QO50 group were significantly higher than those in the other groups (*p* < 0.05). The contents of C22:0 and the SFAs in group Q8 were the highest. The content of C18:2n6c in the QO150 group was the highest. There was no significant difference in the contents of other fatty acids between the Q8 group and the experimental groups (*p* > 0.05).

### 3.2. Effects of Dietary Oregano Essential Oil on Muscle Fatty Acids of Luhua Chicken Legs

The composition of the fatty acids in each group was similar in the leg muscles (Table 3). A total of 27 fatty acids were detected, including 10 SFAs and 17 UFAs. In the UFAs, there were eight species of MUFAs and nine species of PUFAs. Among the 27 fatty acids, the content of C18:1n9c was the highest with about 34%, followed by C16:0, with about 26%, and C18:2n6c with about 15%. It can be seen from the table that some of the fatty acids of the leg muscles were significantly different when different proportions of oregano essential oil were added to the diets. After adding oregano essential oil, the contents of C14:1, C15:0, C15:1, C17:1, C20:1 and C20:2 in the experimental groups (QO50, QO100 and QO150) were significantly lower than those in the Q8 group (*p* < 0.05). The contents of C18:3n3, C20:5n3 (EPA), C22:6n3 (DHA), UFAs and n-3 PUFAs in the QO100 and QO150 groups were significantly lower than those in the Q8 and QO50 groups (*p* < 0.05), while the contents of C16:0, the SFAs and the n-6/n-3 ratio were the opposite (*p* < 0.05). There was no significant difference in the contents of the other fatty acids between the Q8 group and experimental groups (*p* > 0.05).

### 3.3. Cecal Microbial Diversity

A PCR-free library was constructed based on the Illumina Nova sequencing platform, and paired-end sequencing was performed. By splicing Reads, an average of 90,051 tags were measured in each sample (Table 4), and 85,240 suitable data were obtained on average after quality control measures, with an effective data volume of 66,575 and a quality control efficiency of 73.95%. The Uparse software was used to cluster the effective tags at a similarity level of 97% to obtain the OUT number of each sample. A total of 1032 OTUs were obtained(Figure 2A), including 926 OUTs in the Q8 group, 774 OUTs in the QO50 group and 783 OUTs in the QO100 group. The QO150 group had 775 OUTs. The numbers of unique OTUs in the Q8, QO50, QO100 and QO150 groups were 166, 11, 11 and 22, respectively. The unique OUT number in the Q8 group was significantly higher than that in the other groups. The number of OTUs shared by the four groups was 656. The OTU sequence and Silva132 database were annotated for species. In the annotation results, a total of 418 (40.50%) OTUs were annotated to the genus level. The dilution curve described the species diversity and richness of each sample. At 40,000 reads, the curve tended to be gentle, indicating that the sequencing coverage was saturated, the uniformity of the species in the sample tended to be stable under the test conditions, and the current sequencing depth of each sample was sufficient to reflect the microbial diversity contained in the community sample (Figure 2B). As shown in Table 5, the Alpha diversity index showed that there was no significant difference between each group. As shown in the PCA figure (Figure 2C), there were significant differences in the cecal microbial species between the Q8 group and the experimental groups with different amounts of oregano essential oil. Further results of the Anosim analysis were consistent with those of the PCA, and the differences between the groups were greater than those within the groups (Figure 2D). The species accumulation curve (Figure 2E) showed that with the increase in the sample size, the number of species and common species in the environment reached a saturation point and will not increase with an increase in the sample size.

### 3.4. Cecal Microbial Species Composition and Abundance

By forming comparisons with the Silva132 database which annotates the species, a total of 1032 OTUs were found at different classification levels. The number of OTUs that could be annotated in the database was 1025 (99.32%). The proportion of OTUs annotated to the kingdom level was 99.32%, the phylum level was 97.48%, the class level was 95.16% and the order level was 91.47%. The proportion at the family level was 81.69%, at the genus level was 40.50%, and at the species level was 14.27% (Table 6).

At the phylum level, *Bacteroidetes*, *Firmicutes*, *Proteobacteria* and *Synergistetes* were the dominant phyla, with relative abundances of more than 1%. The relative abundance of *Bacteroidetes* and *Firmicutes* was the highest in all the experimental groups, accounting for more than 90% of the total abundance (Figure 3A). At the genus level, there were 30 bacteria genera with relative abundances greater than 0.1%, and *Bacteroides*, *unidentified_Lachnospiraceae* and *Lactobacillus* were the dominant bacteria genera in all the experimental groups (Figure 3B).

### 3.5. Analysis of Significant Differences among Species

At the phylum level (Figure 4), the number of *Proteobacteria* in the QO50 group was significantly higher than that in the QO150 group (*p* = 0.016), and the number of *Elusimicrobia* in the Q8 group was significantly higher than that in the QO150 group (*p* = 0.028). At the genus level (Figure 5), comparing the Q8 group with the QO50 group, the number of *Phascolarctobacterium* in the Q8 group was significantly higher than that in the QO50 group, while the numbers of *Lachnoclostridium*, *unidentified_Christensenellaceae* and *Enorma* in the QO50 group were significantly higher than those in the Q8 group (*p* < 0.05). Comparing the QO50 group with the QO100 group, the number ofs *Bacteroides* and *Enorma* in the QO50 group were significantly higher than those in the QO100 group (*p* < 0.05). The numbers of *Desulfovibrio*, *unidentified_Ruminococcaceae* and *Mailhella* in the QO50 group were significantly higher than in the QO150 group, as determined by a T test. Comparing the Q8 group with the QO100 group, the numbers of *Megamonas*, *Lachnoclostridium*, *Fournierella*, *Erysipelatoclostridium*, *Ruminiclostridium*, *Tyzzerella*, *Enorma*, and *Negativibacillus* in the QO100 group were significantly higher than those in the Q8 group, while the numbers of *Phascolarctobacterium*, *Parasutterella* and *Bilophila* in the QO100 group were significantly lower than those in the Q8 group (*p* < 0.05). Comparing the QO100 group with the QO150 group, the numbers of *Faecalibacterium*, *unidentified_Ruminococcaceae*, *Mailhella* and *Negativibacillus* in the QO100 group were significantly higher than those in the QO150 group (*p* < 0.05). Comparing the Q8 group with the QO150 group, the numbers of *Phascolarctobacterium*, *Elusimicrobium*, *Parasutterella*, *Mailhella* and *Bilophila* in the Q8 group were significantly higher than those in the QO150 group. The numbers of *Ruminiclostridium* and *Enorma* in the Q8 groups were significantly lower than those in the QO150 group (*p* < 0.05).

### 3.6. Prediction of Cecal Microbial Function

We used software to predict the abundance of functional genes with significant differences in the L2 level of the KEGG pathway (*n* = 6/group), and thus analyze the functional differences between the two groups. The analysis’ results are shown in Figure 6. Comparing the Q8 group with the QO50 group, *Amino Acid Metabolism* and *Energy Metabolism* were significantly enriched in the QO50 group (*p* < 0.05). *Poorly Characterized* was significantly enriched in the Q8 group (*p* = 0.034). Comparing the QO50 group with the QO100 group, *Genetic Information Processing* was significantly enriched in the QO100 group (*p* = 0.014). *Metabolism* and *nervous system* were significantly enriched in the QO50 group (*p* < 0.05). Comparing the Q50 group with the QO150 group, *Signal Transduction* was significantly enriched in the QO50 group (*p* = 0.019). Comparing the QO100 group with the Q8 group, *Energy Metabolism* was significantly enriched in the QO100 group (*p* = 0.008). Comparing the QO100 group with the QO150 group, *Metabolism*, *Metabolism of Other Amino Acids*, and *Biosynthesis of Other Secondary metabolites* were significantly enriched in the QO150 group (*p* < 0.05), while *Signal Transduction* was the opposite (*p* = 0.036). Comparing the QO100 group with the QO150 group, *Amino Acid Metabolism*, *Energy Metabolism*, *Metabolism of Other Amino Acids* and *Biosynthesis of Other Secondary metabolites* were significantly enriched in the QO150 group (*p* < 0.05).

### 3.7. Correlation Analysis

There was a significant correlation between the cecal microbiota and muscle fatty acid deposition (Figure 7), and among the fatty acids in the pectoralis muscle (Figure 7A), the abundance of *Bacteroides* was significantly positively correlated with the content of C18:0 and C22:6n3 (DHA); the abundance of *unidentified_Lachnospiraceae* was significantly positively correlated with the content of PUFA; the *Romboutsia* abundance was significantly positively correlated with C16:1 and C18:2n6c; and the *Others* abundance was significantly positively correlated with C18:1n9c and C18:3n3 (*p* < 0.01). The abundance of *Bacteroides* was significantly negatively correlated with the contents of C18:1n9c and C18:3n3, and the abundance of *Others* was significantly negatively correlated with the contents of C18:0 and C22:6n3 (DHA) (*p* < 0.01). In the fatty acids of the leg muscles (Figure 7B), the abundance of *Bacteroides* was significantly positively correlated with the n-6/n-3 ratio; the abundance of *Lactobacillus*, *Lawsonia*, and *Faecalibacterium* was positively correlated with the content of C18:0; the abundance of *Romboutsia* was significantly positively correlated with the content of C16:1; the abundance of *Alistipes*, *Phascolarctobacterium* and *Desulfovibrio* was significantly and positively correlated with the content of C20:3n6, C22:0 and n-3 PUFAs; and the abundance of *Others* was significantly positively correlated with the content of C22:6n3 (DHA) (*p* < 0.01).

## 4. Discussion

### 4.1. Effects of Oregano Essential Oil on Muscle Fatty Acids of Luhua Chicken

The composition and content of muscle fatty acids are key indicators to evaluate the nutritional value of the muscle [31]. Relevant studies have shown that a high concentration of n-3 PUFAs could increase the concentration ratio of unsaturated fatty acids (PUFA/SFA) and n-3/n-6, which are beneficial to human health [32]. From the perspective of nutrition and health, the composition and content of muscle fatty acids are important indicators to measure meat quality [33]. Fatty acids in muscle can form the precursors of flavor substances through relevant reactions; additionally, PUFAs can effectively prevent some chronic diseases, reduce fat, regulate the body’s autogenic immunity, and promote growth and development, which are beneficial to human health [34]. MUFAs can improve memory, reduce blood sugar and regulate blood lipids to a certain extent [35]. The results of this study showed that dietary supplementation of a certain amount of oregano essential oil resulted in the following: the breast muscle SFAs in the experimental groups were lower than those in the Q8 group; the PUFAs in the QO50 and QO100 groups were significantly lower than those in the Q8 group; and the PUFAs in the QO100 group were significantly higher than those in the control group. In leg muscle fatty acids, the contents of C14:1, C15:0, C15:1, C17:1, C20:1 and C20:2 in the experimental groups (QO50, QO100 and QO150) were significantly lower than those in Q8 group; these results indicate that adding a certain proportion of oregano essential oil in the diet can improve the breast muscle UFAs of Luhua chickens to a certain extent. Because of its involvement in the Maillard reaction, it increases the content of flavor compounds, thus increasing the meat flavor [36]. The more abundant the UFAs’ composition and content in muscle, the higher its nutritional value and the more balanced it is. Studies on oregano essential oil in poultry fatty acids have been rarely conducted, and the results of this study are inconsistent with those obtained by Liu Lishan et al. [37] in the study of Holstein bulls, which may be due to the different intestinal ecosystems of monogastric animals and ruminants. The growth-promoting mechanism leading to the digestion, absorption and efficacy of oregano essential oil in the intestinal tract of chickens is different from rumen fermentation and related reactions in ruminants [38]. It is also possible that the addition of oregano essential oil in the diet may affect the cecal microflora environment of chickens, inhibit the number of harmful bacteria, and cause the related anabolism of fatty acids. In this study, adding different proportions of oregano essential oil to the diet increased the content of elaidic acid (C18:1n9t) in breast and leg muscles, and decreased the content of linolenic acid (C18:3n3) in leg muscle, which was consistent with the results of Wood et al. [39]. Meanwhile, studies have found that C18:2n6c can be metabolized to C20:4n6, and C18:3n3 can be metabolized to EPA and DHA, which play an important role in the prevention and treatment of prostaglandins, thrombosis, and atherosclerosis, as well as in immunity, anti-inflammatory and membrane functions [40,41]. It is speculated that the phenolic compounds (carvacrol and thymol) in oregano essential oil have antioxidant, anti-inflammatory and antibacterial properties, which inhibit the activity of hydrogenated microorganisms, and thus affect the synthesis of some fatty acids in muscle. However, the specific reason needs to be further explored. C18:1n9 is the main product of de novo synthetic fat in animals, and its concentration increases with the increase in the IMF content in the animal [31,42]. The n-6/n-3 and PUFA/SFA ratios are key indicators for evaluating the nutritional value of muscle. Among them, the World Health Organization (WHO) recommends the range of the n-6/n-3 ratio to be (5–10):1 and the appropriate PUFA/SFA ratio to be greater than 0.4 [41,43]. Meanwhile, the recommended range of the n-6/n-3 ratio in Reference Intake of Dietary Nutrients for Chinese Residents is (4–6):1 [44]. At present, the optimal range of this ratio is still controversial. The results of this study showed that the ratios of n-6/n-3 and PUFA/SFAs in breast muscle fatty acids were 7.0 and 0.55, respectively. The ratios of n-6/n-3 and PUFA/SFAs in leg muscle fatty acids were 6.0 and 0.6, respectively. The ratios of n-6/n-3 and PUFA/SFAs are all within the suitable range recommended by the WHO, suggesting that dietary supplementation of different proportions of oregano essential oil will not have adverse effects on the muscle fatty acids of broilers, and to some extent, it can also improve the fatty acid content in the muscles of Luhua chickens. However, the regulation mechanism of oregano essential oil on fatty acids needs to be further studied.

### 4.2. Effects of Oregano Essential Oil on Cecal Microflora of Luhua Chickens

Currently, intestinal microorganisms are receiving increasing attention due to their vital role in intestinal development and metabolic homeostasis [45]. The small intestine, colon, and cecum have similar roles in the digestion and absorption of nutrients. The cecum of chickens is the most important distal intestinal segment of chickens, and the intestinal microbe concentration is the highest in adult chickens. The number of microbes is significantly higher than in other intestinal segments in poultry with a diverse microbiota, which affects the health and growth performance of the body [14,46]. Therefore, it is necessary to study the intestinal microbial diversity of Luhua chickens.

The alpha diversity can reflect the diversity and richness of microbial colonies, which are measured by Observed_species, Shannon, Simpson, Chao1, ACE and other indexes. The results of this study showed that the indexes of Observed_species, Shannon, Simpson, Chao1, ACE, and PD_whole_tree in the experimental groups were slightly lower than those in the Q8 group when different concentrations of oregano essential oil were added in the diet, but the differences were not significant. These results indicated that the concentration of oregano essential oil added in this study had no significant effect on the diversity and richness of the cecal microbial colonies of the broilers. This may be caused by the fact that oregano essential oil has certain antibacterial activities against intestinal microorganisms, such as *Escherichia coli* and *Clostridium perfringens* [47,48]. The antibacterial properties of essential oils in poultry may be affected by their basal diets and environmental conditions [5]. PCA and NMDS analysis methods of the beta diversity were used to evaluate the differences in the species diversity among the different cecal microbial samples. The results of this study showed that there were significant differences between the Q8 group and the experimental groups supplemented with different concentrations of oregano essential oil. The microbiota of the control group, the Q8 group and the experimental groups supplemented with different concentrations of oregano essential oil could be significantly distinguished, but the microbiota of the experimental groups supplemented with different concentrations of oregano essential oil could not be distinguished. These results indicated that there were some differences in the microbial structure between the Q8 group and the experimental groups supplemented with different concentrations of oregano essential oil. *Firmicutes* were dominant in the chickens’ cecum microflora [49,50]. Yan et al. [51] found in their study that *Bacteroidetes* accounted for the highest proportion of cecal microbes in chickens, which reached 50%, while *Firmicutes* accounted for only 20%. Related studies have also found that *Bacteroides*, *Firmicutes* and *Proteobacteria* are the dominant bacterial communities in the cecal microflora of poultry [49,51] and marine mammals [52]. In this study, at the phylum level, *Bacteroidetes*, *Firmicutes*, *Proteobacteria* and *Synergistetes* were the dominant phyla, with relative abundances of more than 1%. The relative abundance of *Bacteroidetes* and *Firmicutes* was the highest out of all the experimental groups, accounting for more than 90% of the total abundance. This result is similar to that of Dong et al. [53]. *Firmicutes* and *Bacteroidetes* play important roles in the energy metabolism of animals [54]: *Firmicutes* can produce various digestive enzymes to promote the digestion and absorption of nutrients [55], and *Bacteroidetes* have shown a strong degradation of protein and carbohydrates [56,57]. The greater the ratio of *Firmicutes*:*bacteroidetes* in the gut, the greater the absorption capacity of the body for energy-related substances; the smaller the ratio, the more prone the animal becomes to obesity [56,58]. *Bacteroidetes* and *Firmicutes* constitute most of the microbial communities at the level of intestinal microbiota in broilers. The results of this study are consistent with those of that study. The relative abundance of *Firmicutes* and *Bacteroidetes* is the highest in all the experimental groups. Huang et al. [59] found that oregano essential oil increased the number of *Firmicutes* and decreased the number of *Actinobacteria* and *Proteobacteria*. With the increase in *Firmicutes*, the inflammatory response was weakened and the intestinal barrier functioning was enhanced. At the genus level, there were 30 genera with a relative abundance greater than 0.1%, and *Bacteroides*, *unidentified_Lachnospiraceae* and *Lactobacillus* were the dominant genera in all the experimental groups. Studies have found that *Lactobacillus* can promote the development of the intestinal tract, maintain the integrity of intestinal morphology, and play a certain role in promoting immune regulation [60]. *Bacteroidetes* can participate in the synthesis of short-chain fatty acids [61], which can significantly improve the intestinal environment of animals, regulate the electrolyte balance and improve the metabolic function of the intestinal tract of animals [49,51,52]. *Bacteroides* in the QO100 group were significantly higher than those in the other groups, indicating that 100 mg·kg^−1^ oregano essential oil added in the diet was conducive to the fermentation of *Bacteroides* in the cecum, the production of acetate, propionate and other short-chain fatty acids, and the maintenance of intestinal health homeostasis. Dong R. et al. [11] showed that adding a certain amount of oregano essential oil in diets increases the relative abundance of *Firmicutes* and *Lactobacillus*. *Firmicutes* represent the largest population of gut microbes in mice and humans [62], and many species of this phylum are capable of producing endospores that are resistant to drying and other harsh conditions [63]. This is consistent with the results of this study. These results suggest that adding a certain amount of oregano essential oil in the diet can improve the intestinal health and metabolism of animals.

Using the PICRUST software to predict KEGG function, this study found that *Amino Acid Metabolism*, *Energy Metabolism*, *Metabolism*, *Nervous System* and *Signal Transduction* were significantly enriched in the QO50 group, *Genetic Information Processing* was significantly enriched in the QO100 group, *Metabolism of Other Amino Acids* and *Biosynthesis of Other Secondary Metabolites* were significantly enriched in the QO150 group, and *Poorly Characterized* was significantly enriched in the Q8 groups. This result is similar to that of Choi et al. [64], suggesting that dietary oregano essential oil with a certain concentration may change the community composition and microbial function of the cecum of broilers. The *Amino Acid Metabolism*, *Energy Metabolism*, *Metabolism*, *Signal Transduction*, *Genetic Information Processing* pathways can promote the digestion and absorption of nutrients and enhance intestinal barrier functioning.

### 4.3. Correlation Analysis

Intramuscular fat deposition in animals is regulated by intestinal flora [65]. The abundance of *Bacteroides* was positively correlated with the immune system, glycan biosynthesis and metabolism, as well as the signaling pathways related to cofactors and vitamin metabolism. The genus *Bacteroides* exhibits a high degree of host specificity, reflecting individual differences in the digestive systems of the host animals [66]. The correlation analysis between cecal microbes and muscle fatty acids showed that there was a significant correlation between cecal microbes and muscle fatty acid deposition. In breast muscle, the abundance of *Bacteroides* was positively correlated with the contents of C18:0 and C22:6n3 (DHA), and *Romboutsia* was positively correlated with the contents of C16:1 and C18:2n6c. The abundance of *Bacteroides* was negatively correlated with the contents of C18:1n9c and C18:3n3. In leg muscles, the abundance of *Bacteroides* and the ratio of n-6/n-3, the abundance of *Lactobacillus*, *Lawsonia* and *Faecalibacterium* and the content of C18:0, the *Romboutsia* abundance and the C16:1 content, the *Alistipes*, *Phascolarctobacterium*, *Desulfovibrio* abundance and C20:3n6, C22:0 and n-3 PUFA content all have extremely significant positive correlations. This is mainly because *Bacteroides* and *Lactobacillus* are the main bacteria of fatty acid hydrogenation, and *Faecalibacterium* co-occurs with several members of *Bacteroidetes*. *Bacteroidetes* may be the main bacteria of linolenic acid (C18:3n3) hydrogenation [67]. Because of *bacteroides*, *lactobacillus* contributes to the changes in abundance in the cecum and leads to changes in the abundance of other bacterial genera in the cecum. This indicates that there is a synergistic effect among the various microbial genera in the caecum, thus affecting muscle fatty acid deposition and leading to the change in the fatty acid content in muscles. However, the exact regulatory mechanism is not clear, and needs to be further explored.

## 5. Conclusions

Dietary oregano essential oil can effectively increase the contents of elaidic acid (C18:ln9t), polyunsaturated fatty acids (PUFAs) and n-3 polyunsaturated fatty acids (n-3 PUFAs) in breast muscle tissues. At the genus level, the amounts of *Phascolarctobacterium*, *Parasutterella* and *Bilophila* in the experimental groups (the QO50, QO100 and QO150 groups) were significantly decreased. The microbial functional enrichment analysis revealed that *Amino Acid Metabolism*, *Energy Metabolism*, *Metabolism*, *Signal Transduction*, *Genetic Information Processing* were mainly enriched in each experimental group. In addition, there was a significant correlation between cecal microorganisms and fatty acid deposition in some muscles. In conclusion, the dietary addition of oregano essential oil can effectively improve the fatty acid content of breast muscles and the composition of cecal microbial communities, promote nutrient digestion and absorption, and enhance the intestinal barrier functioning of Luhua chickens.

## Figures and Tables

**Figure 1 animals-12-03215-f001:**
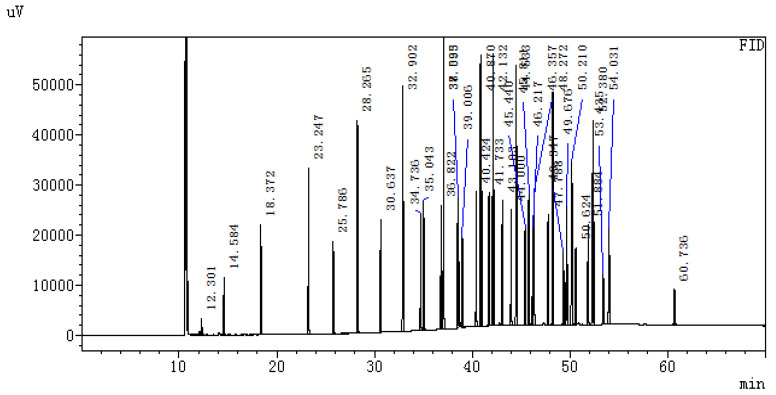
Peak diagram of standard sample.

**Figure 2 animals-12-03215-f002:**
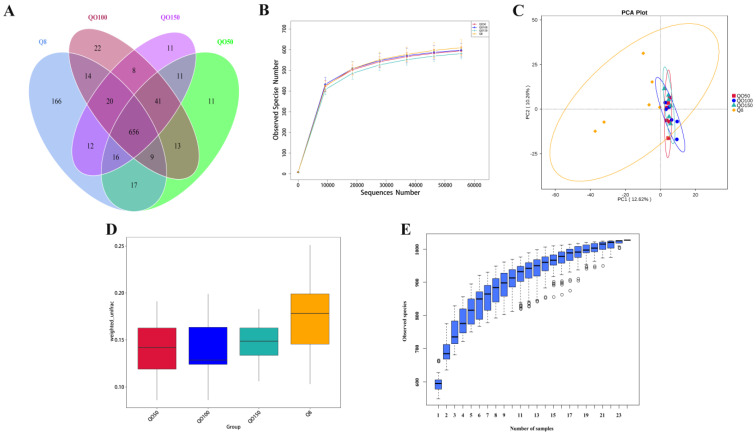
Analysis of cecal microbial diversity. Notes: (**A**) Operational taxonomic units−Venn (OTU−Venn) analysis; (**B**) Saturation curve of species number (dilution curve); (**C**) PCA analysis; (**D**) Anosim analysis; (**E**) Species accumulation box diagram.

**Figure 3 animals-12-03215-f003:**
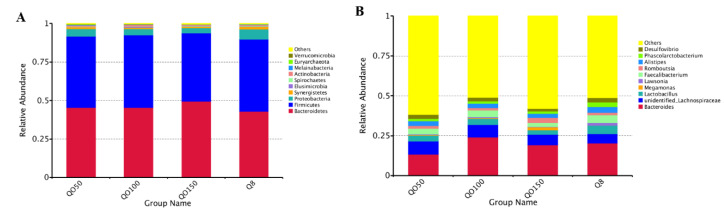
Distribution of microbial abundance. Note: (**A**) Microbe abundance distribution at phylum level; (**B**) Abundance distribution of genus-level microorganisms.

**Figure 4 animals-12-03215-f004:**
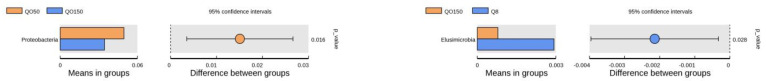
Analysis of species difference between phylum-level T−test groups.

**Figure 5 animals-12-03215-f005:**
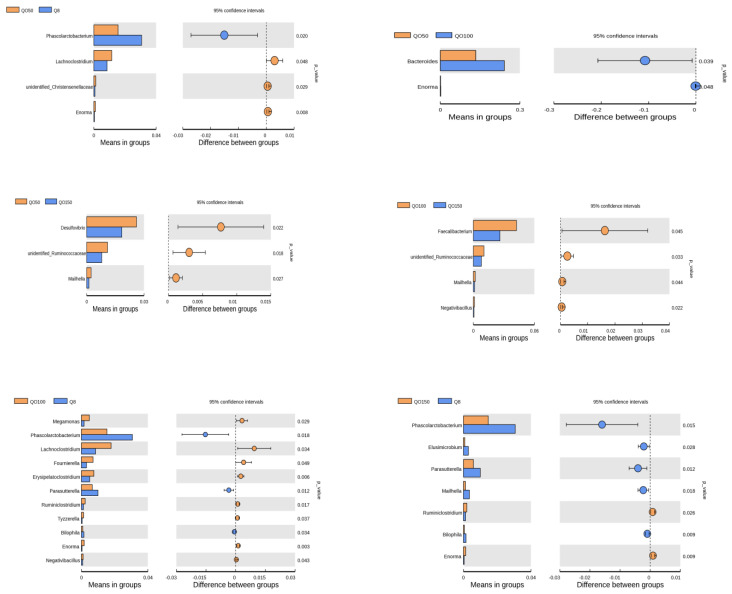
Analysis of species differences between genus-level T−test groups.

**Figure 6 animals-12-03215-f006:**
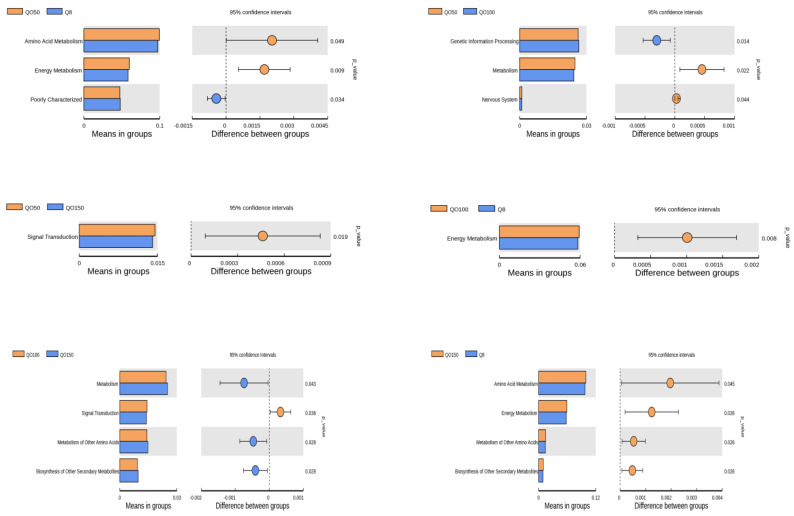
Prediction of cecal microbial function.

**Figure 7 animals-12-03215-f007:**
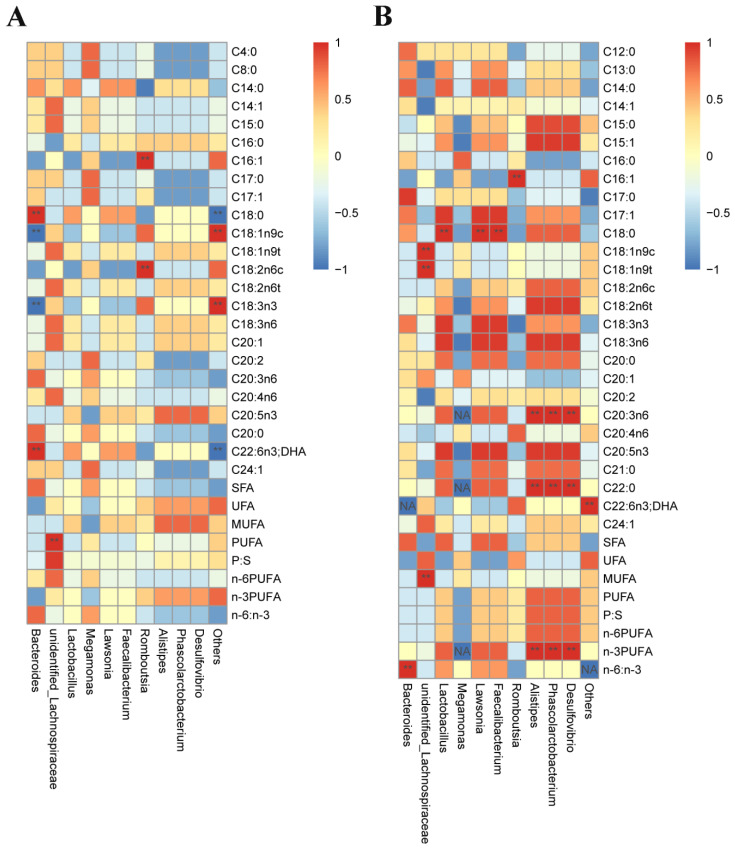
Spearman correlation analysis between cecal microbes and muscle fatty acids at genus level. Notes: (**A**) Correlation analysis between pectoral fatty acids and cecum microbes; (**B**) Correlation analysis between leg muscle fatty acids and cecum microbes. ** *p* < 0.01; ^NA^ No correlation.

**Table 1 animals-12-03215-t001:** Composition and nutrient level of experimental diets (dry matter basis).

Items	Diets (%)			
	Q8	QO50	QO100	QO150
Corn	64.00	64.00	64.00	64.00
Wheat middling	4.00	4.00	4.00	4.00
Quinoa seeds	8.00	8.00	8.00	8.00
Soybean meal	20.00	20.00	20.00	20.00
Limestone	1.20	1.20	1.20	1.20
CaHPO_4_	1.20	1.20	1.20	1.20
NaCl	0.30	0.30	0.30	0.30
1% Premix ^(1)^	1.00	1.00	1.00	1.00
Oregano oil		0.005	0.01	0.015
zeolite powder	0.30	0.295	0.29	0.285
Total	100	100	100	100
Nutrient levels				
Metabolic energy, ME (MJ/Kg)	12.40	12.40	12.40	12.40
Dry matter, DM	84.66	84.66	84.66	84.66
Crude protein, CP	16.25	16.25	16.25	16.25
Crude fibre, CF	2.74	2.74	2.74	2.74
Ether extract, EE	3.10	3.10	3.10	3.10
Calcium, Ca	0.88	0.88	0.88	0.88
Total phosphorus, TP	0.58	0.58	0.58	0.58
Available phosphorus, AP	0.20	0.20	0.20	0.20
Lysine, Lys	0.84	0.84	0.84	0.84
Methionine, Met	0.33	0.33	0.33	0.33
Methionine + Cystine, Met + Cys	0.61	0.61	0.61	0.61
Threonine, Thr	0.62	0.62	0.62	0.62
Tryptophan, Trp	0.18	0.18	0.18	0.18

Note: Premix (per kilogram of feed content): vitamin A, 158,000 IU; vitamin D, 44,000 IU; vitamin E, 500 IU; vitamin K, 39 mg; vitamin B1, 45 mg; vitamin B2, 125 mg; vitamin B8, 19 mg; vitamin B12, 0.24 mg; niacinamide, 790 mg; pantothenic acid, 300 mg; folic acid, 9 mg; H_2_ O ≤ 8%; D-biotin, 2.9 mg; Zn, 730 mg; Fe, 1400 mg; Mn, 980 mg; Cu, 240 mg; I, 14 mg; Se, 4.8 mg; Ca, 10%; NaCl, 5~13%; Total phosphorus(TP), 3.5%; choline chloride, 7000 mg. Nutrient level is calculated value.

**Table 2 animals-12-03215-t002:** Effects of oregano essential oil on fatty acid composition and content of breast muscle of Luhua chickens (%).

Items	Group				*p*-Values
	Q8	QO50	QO100	QO150	
Butyric acid (C4:0)	0.18 ± 0.06 ^b^	0.32 ± 0.06 ^a^	0.31 ± 0.07 ^a^	0.32 ± 0.09 ^a^	<0.001
Octanoic acid (C8:0)	0.16 ± 0.03	0.21 ± 0.07	0.22 ± 0.06	0.20 ± 0.05	0.277
Myristic acid (C14:0)	0.64 ± 0.09	0.64 ± 0.06	0.58 ± 0.11	0.60 ± 0.06	0.152
Myristoleic acid (C14:1)	0.21 ± 0.03 ^c^	0.23 ± 0.03 ^b^	0.35 ± 0.04 ^a^	0.22 ± 0.03 ^bc^	<0.001
Pentadecanoic acid (C15:0)	2.83 ± 0.62	3.06 ± 0.56	2.87 ± 0.88	2.67 ± 0.82	0.602
Palmitic acid (C16:0)	24.88 ± 3.03 ^b^	24.34 ± 0.87 ^b^	24.43 ± 0.79 ^b^	26.27 ± 1.24 ^a^	0.004
Palmitoleic acid (C16:1)	5.46 ± 0.92	6.11 ± 1.45	5.47 ± 1.02	5.56 ± 0.63	0.182
Heptadecanoic acid (C17:0)	0.41 ± 0.10	0.37 ± 0.09	0.37 ± 0.10	0.31 ± 0.10	0.810
Heptadecenoic acid (C17:1)	0.69 ± 0.06	0.70 ± 0.24	0.71 ± 0.17	0.80 ± 0.17	0.460
Stearic acid (C18:0)	8.40 ± 1.21	8.01 ± 1.00	8.24 ± 0.79	8.00 ± 0.88	0.543
Oleic acid (C18:1 n9c)	31.70 ± 4.22	32.35 ± 3.83	31.25 ± 3.70	31.63 ± 3.28	0.839
Elaidic acid (C18:1n9t)	0.49 ± 0.07 ^d^	1.74 ± 0.08 ^a^	1.53 ± 0.08 ^c^	1.67 ± 0.09 ^b^	<0.001
Linoleic acid (C18:2n6c)	12.23 ± 1.39 ^b^	12.35 ± 0.78 ^ab^	12.68 ± 0.85 ^ab^	13.02 ± 1.21 ^a^	0.032
Linolelaidic acid (C18:2n6t)	0.13 ± 0.01 ^b^	0.16 ± 0.01 ^a^	0.14 ± 0.01 ^b^	0.09 ± 0.002 ^c^	<0.001
Linolenic acid (C18:3n3)	0.69 ± 0.08	0.74 ± 0.17	0.69 ± 0.06	0.75 ± 0.08	0.164
γ-linoleic acid (C18:3n6)	0.16 ± 0.03	0.18 ± 0.08	0.17 ± 0.02	0.14 ± 0.02	0.699
Eicosenoic acid (C20:1)	0.42 ± 0.04	0.43 ± 0.09	0.36 ± 0.12	0.37 ± 0.09	0.113
Eicosadienoic acid (C20:2)	0.45 ± 0.04 ^b^	0.56 ± 0.04 ^a^	0.38 ± 0.09 ^c^	0.41 ± 0.07 ^bc^	<0.001
Eicostrienoic acid (C20:3n6)	0.56 ± 0.13	0.64 ± 0.11	0.53 ± 0.14	0.52 ± 0.16	0.302
Arachidonic acid (AA,C20:4n6)	4.76 ± 1.53	5.84 ± 1.52	5.65 ± 1.65	4.79 ± 1.52	0.133
Eicosapentaenoic acid(EPA,C20:5n3)	1.07 ± 0.12 ^b^	1.19 ± 0.23 ^a^	1.18 ± 0.09 ^a^	1.16 ± 0.12 ^a^	0.039
Behenic acid (C22:0)	0.36 ± 0.07 ^a^	0.30 ± 0.07 ^b^	0.36 ± 0.08 ^ab^	0.36 ± 0.08 ^ab^	0.045
Docosahexaenoic acid (DHA, C22:6n3)	0.67 ± 0.14	0.69 ± 0.22	0.67 ± 0.14	0.70 ± 0.14	0.939
Nervonic acid (C24:1)	1.55 ± 0.26	1.71 ± 0.39	1.66 ± 0.30	1.75 ± 0.35	0.466
Saturated fatty acid (SFA)	38.55 ± 1.62 ^a^	36.75 ± 2.05 ^c^	36.92 ± 1.80 ^bc^	38.13 ± 1.82 ^ab^	0.009
Unsaturated fatty acid (UFA)	62.50 ± 4.65	63.35 ± 2.32	63.24 ± 2.14	61.15 ± 3.28	0.153
Monounsaturated fatty acids (MUFAs)	42.47 ± 6.69	43.07 ± 4.61	41.31 ± 4.44	40.56 ± 4.15	0.429
Polyunsaturated fatty acids (PUFAs)	20.03 ± 2.48 ^b^	20.28 ± 2.40 ^ab^	21.93 ± 2.39 ^a^	20.58 ± 3.18 ^ab^	0.041
n-3 Polyunsaturated fatty acids (n-3 PUFAs)	2.45 ± 0.24	2.51 ± 0.49	2.53 ± 0.14	2.54 ± 0.44	0.876
n-6 Polyunsaturated fatty acids (n-6 PUFAs)	17.44 ± 1.93	17.44 ± 2.47	19.01 ± 2.37	17.92 ± 2.48	0.150
n-6PUFA/n-3PUFA	7.12 ± 0.45 ^ab^	6.67 ± 1.08 ^b^	7.50 ± 0.79 ^a^	7.04 ± 0.50 ^ab^	0.019
PUFA/SFA	0.54 ± 0.06 ^b^	0.55 ± 0.04 ^b^	0.59 ± 0.04 ^a^	0.53 ± 0.09 ^b^	0.015

Note: ^a, b, c^ Values with different superscripts in the same row are significantly different (p < 0.05). The same below.

**Table 3 animals-12-03215-t003:** Effects of oregano essential oil on fatty acid composition and content of Luhua chicken leg muscle %.

Items	Group				*p*-Values
	Q8	QO50	QO100	QO150	
Lauric acid (C12:0)	0.04 ± 0.003	0.10 ± 0.11	0.05 ± 0.02	0.04 ± 0.003	0.246
Tridecylic acid (C13:0)	0.37 ± 0.02 ^b^	0.58 ± 0.08 ^a^	0.52 ± 0.07 ^a^	0.41 ± 0.12 ^b^	<0.001
Myristic acid (C14:0)	0.63 ± 0.11 ^b^	0.67 ± 0.06 ^ab^	0.74 ± 0.06 ^a^	0.72 ± 0.09 ^a^	0.002
Tetradecenoic acid (C14:1)	0.44 ± 0.03 ^a^	0.32 ± 0.04 ^b^	0.20 ± 0.03 ^d^	0.24 ± 0.06 ^c^	<0.001
Pentadecanoic acid (C15:0)	0.10 ± 0.01 ^a^	0.08 ± 0.01 ^b^	0.08 ± 0.01 ^b^	0.08 ± 0.01 ^b^	0.002
Pentadecenic acid (C15:1)	0.35 ± 0.05 ^a^	0.27 ± 0.04 ^b^	0.12 ± 0.02 ^c^	0.15 ± 0.05 ^c^	<0.001
Palmitic acid (C16:0)	26.06 ± 1.29 ^b^	25.78 ± 2.14 ^b^	27.20 ± 0.92 ^a^	27.43 ± 1.48 ^a^	0.002
Palmitoleic acid (C16:1)	7.63 ± 1.02 ^a^	6.56 ± 1.46 ^b^	6.47 ± 0.55 ^b^	7.18 ± 0.99 ^ab^	0.004
Heptadecanoic acid (C17:0)	0.12 ± 0.02 ^ab^	0.11 ± 0.02 ^b^	0.13 ± 0.03 ^a^	0.12 ± 0.01 ^ab^	0.023
Heptadecenoic acid (C17:1)	0.28 ± 0.02 ^a^	0.15 ± 0.03 ^b^	0.16 ± 0.02 ^b^	0.15 ± 0.02 ^b^	<0.001
Stearic acid (C18:0)	7.19 ± 1.02	7.44 ± 0.90	7.54 ± 0.48	7.16 ± 1.12	0.598
Oleic acid (C18:1n9c)	33.12 ± 2.80	35.02 ± 2.70	34.51 ± 1.01	33.63 ± 3.07	0.166
Elaidic acid (C18:1n9t)	0.42 ± 0.03 ^c^	0.61 ± 0.09 ^a^	0.56 ± 0.1 ^b^	0.44 ± 0.04 ^c^	<0.001
Linoleic acid (C18:2n6c)	14.93 ± 1.13	14.75 ± 1.12	14.77 ± 0.67	15.27 ± 1.33	0.477
Linolelaidic acid (C18:2n6t)	0.09 ± 0.02	0.09 ± 0.03	0.06 ± 0.01	0.09 ± 0.14	0.933
Linolenic acid (C18:3n3)	1.38 ± 0.07 ^a^	1.37 ± 0.11 ^a^	1.28 ± 0.05 ^b^	1.20 ± 0.08 ^c^	<0.001
γ-linoleic acid (C18:3n6)	0.19 ± 0.04 ^a^	0.17 ± 0.03 ^a^	0.17 ± 0.02 ^ab^	0.15 ± 0.03 ^b^	0.003
Arachidic acid (C20:0)	0.05 ± 0.01	0.05 ± 0.004	0.08 ± 0.02	0.04 ± 0.009	0.173
Eicosenoic acid (C20:1)	0.41 ± 0.05 ^a^	0.22 ± 0.03 ^b^	0.26 ± 0.09 ^b^	0.22 ± 0.03 ^b^	<0.001
Eicosadienoic acid (C20:2)	0.16 ± 0.03 ^a^	0.12 ± 0.03 ^b^	0.12 ± 0.02 ^b^	0.11 ± 0.03 ^b^	<0.001
Eicostrienoic acid (C20:3n6)	0.36 ± 0.1	0.34 ± 0.12	0.30 ± 0.04	0.34 ± 0.14	0.567
Arachidonic acid (AA, C20:4n6)	2.18 ± 0.46	2.16 ± 0.58	2.43 ± 0.55	2.14 ± 0.30	0.876
Eicosapentaenoic acid (EPA, C20:5n3)	0.92 ± 0.07 ^a^	0.90 ± 0.09 ^a^	0.80 ± 0.13 ^b^	0.74 ± 0.09 ^b^	<0.001
Heneicosan oic acid (C21:0)	0.04 ± 0.002	0.04 ± 0.003	0.04 ± 0.004	0.04 ± 0.005	0.434
Behenic acid (C22:0)	0.15 ± 0.05	0.17 ± 0.06	0.16 ± 0.04	0.15 ± 0.04	0.913
Docosahexaenoic acid (C22:6n3, DHA)	0.84 ± 0.03 ^a^	0.85 ± 0.1 ^a^	0.58 ± 0.05 ^b^	0.57 ± 0.08 ^b^	<0.001
Nervonic acid (C24:1)	0.60 ± 0.12 ^a^	0.64 ± 0.18 ^a^	0.54 ± 0.09 ^ab^	0.45 ± 0.17 ^b^	0.001
Saturated fatty acid (SFA)	34.76 ± 1.43 ^b^	34.93 ± 2.38 ^b^	36.43 ± 0.90 ^a^	36.34 ± 1.31 ^a^	0.001
Unsaturated fattyacid (UFA)	65.22 ± 1.44 ^a^	65.07 ± 2.38 ^a^	63.57 ± 0.90 ^b^	63.70 ± 1.33 ^b^	0.001
Monounsaturated fatty acids (MUFA)	43.24 ± 2.93 ^b^	65.07 ± 2.38 ^a^	42.83 ± 0.94 ^b^	42.44 ± 3.79 ^b^	<0.001
Polyunsaturated fatty acids (PUFA)	21.98 ± 2.38	21.35 ± 1.83	20.74 ± 0.88	21.26 ± 3.50	0.550
n-3 polyunsaturated fatty acids (n-3 PUFAs)	3.14 ± 0.13 ^a^	3.12 ± 0.21 ^a^	2.51 ± 0.27 ^b^	2.51 ± 0.18 ^b^	<0.001
n-6 polyunsaturated fatty acids (n-6 PUFAs)	18.69 ± 2.35	18.13 ± 1.73	18.05 ± 0.83	18.67 ± 3.35	0.811
n-6 PUFA/n-3 PUFA	5.96 ± 0.75 ^b^	5.82 ± 0.50 ^b^	6.86 ± 0.27 ^a^	6.71 ± 0.33 ^a^	<0.001
PUFA/SFA	0.63 ± 0.07 ^a^	0.61 ± 0.03 ^ab^	0.57 ± 0.03 ^b^	0.59 ± 0.1 ^ab^	0.044

**Table 4 animals-12-03215-t004:** Data preprocessing statistics and quality control.

Items	Sample Group				*p*-Values
	Q8	QO50	QO100	QO150	
Raw PE	91,771 ± 6326	91,706 ± 4162	86,699 ± 5442	90,030 ± 5086	0.335
Raw Tags	88,060 ± 6492	86,349 ± 5064	81,734 ± 5208	84,817 ± 4592	0.247
Clean Tags	86,507 ± 6438	84,766 ± 4912	80,095 ± 5134	83,171 ± 4487	0.223
Effective Tags	67,843 ± 4736	68,198 ± 3673	62,426 ± 3300	67,831 ± 4192	0.063
Base	28,334,530 ± 1,964,222	28,425,931 ± 1,558,632	26,017,239 ± 1,374,162	28,328,611 ± 1,802,962	0.061
AvgLen	417.5 ± 1.05	416.83 ± 0.41	416.67 ± 1.03	417.67 ± 1.21	0.240
Q20	98.45 ± 0.09	98.405 ± 0.14	98.35 ± 0.06	98.35 ± 0.04	0.191
Q30	94.94 ± 0.22	94.84 ± 0.35	94.72 ± 0.12	94.71 ± 0.08	0.276
GC (%)	53.22 ± 0.41	53.26 ± 0.15	52.94 ± 0.24	53.40 ± 0.35	0.100
Effective (%)	73.98 ± 3.37	74.375 ± 2.47	72.085 ± 2.94	75.35 ± 2.15	0.256

**Table 5 animals-12-03215-t005:** Alpha diversity indices.

Items	Group				*p*-Values
	Q8	QO50	QO100	QO150	
observed_species	609.67 ± 42.41	594.17 ± 21.57	597.50 ± 36.73	581.00 ± 27.46	0.528
Shannon	6.16 ± 0.18	6.15 ± 0.42	6.29 ± 0.38	5.97 ± 0.29	0.426
Simpson	0.96 ± 0.01	0.95 ± 0.03	0.96 ± 0.02	0.95 ± 0.01	0.572
chao1	643.82 ± 44.96	621.40 ± 22.59	626.14 ± 35.73	616.97 ± 27.55	0.547
ACE	647.60 ± 44.69	623.57 ± 20.05	624.87 ± 33.66	615.07 ± 24.75	0.364
PD_whole_tree	39.11 ± 2.25	40.94 ± 3.65	41.12 ± 4.65	38.61 ± 3.05	0.516

**Table 6 animals-12-03215-t006:** Species annotation analysis.

OTU catalogue	1032	Annotated on Class level:	95.16%
Annotated on (database)	1025 (99.32%)	Annotated on Order level	91.47%
Annotated on Unclassified	7 (0.68%)	Annotated on Family level	81.69%
Annotated on Kingdom level	99.32%	Annotated on Genus level	40.50%
Annotated on Phylum level	97.48%	Annotated on Species level	14.24%

## Data Availability

The sequencing data were deposited into the Sequence Read Archive (SRA) of NCBI (PRJNA903275).

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
