# Peer review of "Effects of Dietary Oregano Essential Oil on Cecal Microorganisms and Muscle Fatty Acids of Luhua Chickens"

_animals, 2022, doi:10.3390/ani12223215_

Round 1

Reviewer 1 Report

The research is good, but it is better to experiment with commercial strains

Reviewer 2 Report

The authors evaluated the effects of a dietary supplementation with oregan essential on cecal microbiota and muscle fatty acids in Luhua Chickens. In particular, they used different concentration of this additive compared to animals with a basal diet. Oregano essential oils seems to have a good influence on treated chickens. 

Even thought there was conducted a great amount of experiment this manuscript has major shortcomings, mostly in the experimental design and the exposed results. 

Here my comments:

Table 1: Why you did not perform the analysis of fatty acids and total lipid?

Line 125: Materials and methods. I could not find in this chapter the information about condition of the rearing house, the temperature, humidity and hours of light. Where the chickens where allotted? In boxes? Dimension of boxes?

Line 161: How many days before the experiment the chicken house was disinfected? How big is the house? It is a large shed?

Line 162: Immunized? How? Did you treat the chickens? Please specify

Line 163: How do you disinfected the chickens? And what do you mean for disinfection? Please specify

Line 163: the pigs?

Line 164: This is not clear? The feed was administered ad libitum, or it was calculated the daily amount? If yes, how do you do it? Since there are not data of daily feed consumption, feed intake and average daily gain?

Lines 166-167: Reformulate, it is not clear. Experiment in 75 or 73 days?

Line 170: If you selected the chickens basing on good health you compromise the experiment. The best condition could have been bringing half of the group in slaughtering house and choose the breast and legs randomly in slaughtering house basing on each group.

Line 172: hair? Feathers

Line 177: Again, the selection is not scientifically accurate.

Table 2, 3, 4, 5: Where are the corresponded p-values?

Line 219: I would write total DNA instead of genomic DNA.

Lines 527-529: how you can state it, if you are evaluating only the cecal microbiota?

Reviewer 3 Report

I read this manuscript for possible publication in Animals. This paper deals with effects of dietary oregano essential oil on cecal microorganisms and muscle fatty acids of Luhua chickens. The manuscript written within the scope of the journal. The studies were carried out in according to well-proven methods, but I can’t understand why 49 day old chickens were included in the study? Typically, for this type o study, one-day-old chicks are used. Moreover, the experiment was ended on the 121st day of age. Very late for chickens! The authors of the manuscript also mentioned in the abstract about the sequencing of the bacterial 16S rRNA. However, in the methodology, I did not find a description of how it was performed?

Another point – why genomic DNA was extracted by CTAB methods? This method is most often used to extract DNA from plants, to disrupt their membranes.

My basic remark also applies to the English language, which is not at the best level and also not the best style, punctuation errors do not make it easier to read this manuscript.

The abstract is a bit too long. I would change the order of the keywords.

The same passages repeat in the introduction.

I would change the subheadings in the materials and methods.

Besides, in the materials and methods, when describing feeding management, there is a passage about feeding pigs!!! This shows that this passage was copied from another article.

Then we have an excerpt: “the chickens were bled to death by neck, hair was removed....” Hair? I guess there should be feathers????

Why were the muscles collected earlier than the intestines?

The layout of the manuscript is not entirely well received.

In the results and discussion there are many naming errors and this part of manuscript is not so good described. Not accurate descriptions of the results. The results are difficult to understanding.

What is CK group? Should be probably Q8 group?

In discussion, there are few references to other studies.

Conclusion need to be improved.

The table titles are not correct.

This manuscript in such form should be considered as not acceptable for publication in Animals and alternatively it needs major revisions!

Round 2

Reviewer 2 Report

I appreciate very much the effort of authors in changing and correcting the manuscript following my suggestion. Now is suitable for pubblication, that I strongly reccomend. 

Author Response

Many thanks to the reviewers for their review of the manuscript and their valuable suggestions for revision .

Reviewer 3 Report

I read this manuscript again for possible publication in Animals. Yes, the manuscript has been improved, but only slightly, and still contains parts that need to be corrected.

My suggestions again:

Ln 42: lack dots after sentence
Ln 60: it is not clear
Ln 64-66: this sentence needs to be changed
Ln 67-71: this passage is a repetition of what was previously mentioned
Ln 87-90: this passage needs to be rewritten
Ln 93: after "diet" needs to be dot
Ln 112-115: this sentence needs to be changed
Ln 139: "it was composed of..." what was? not clear
2.3.2 - that's not a good title
Ln 277-278: it is not necessary because it is methodology
Ln 321: after "775 OUTs needs to be dot
Ln 324: after Figure there should be no dot
Ln 325: after 418 there should be a space
Ln 487: change that sentence
Ln 595: after group should be dot

The subheadings and layoutof text in the Material and methods have not been changed. Practically no text modification.Similarly, no changes to the description of the results and discussions. Table titles changed imperceptibly. English has not been fully corrected and there are still stylistic errors.

The manuscript needs further corrections.
